# The risk of Long Covid symptoms: a systematic review and meta-analysis of controlled studies

Lauren L. O'Mahoney [1,20], Ash Routen [1,20], Clare Gillies[1,2], Sian A. Jenkins[3], Abdullah Almaqhawi [4], Daniel Ayoubkhani [2,5], Amitava Banerjee [6,7], Chris Brightling[8], Melanie Calvert [9,10,11], Shabana Cassambai[1], Winifred Ekezie[12], Mark P. Funnell [1], Anneka Welford[1], Arron Peace[1], Rachael A. Evans [8,13], Shavez Jeffers [1], Andrew P. Kingsnorth[14], Manish Pareek[8,15], Samuel Seidu[1], Thomas J. Wilkinson[1], Andrew Willis[16], Roz Shafran [17], Terence Stephenson [17], Jonathan Sterne [18], Helen Ward [19], Tom Ward [8] & Kamlesh Khunti [1] ✉

The global evidence on the risk of symptoms of Long Covid in general populations infected with SARS-CoV-2 compared to uninfected comparator/control populations remains unknown. We conducted a systematic literature search using multiple electronic databases from January 1, 2022, to August 1, 2024. Included studies had ≥100 people with confirmed or self-reported COVID-19 at ≥28 days following infection onset, and an uninfected comparator/control group. Results were summarised descriptively and meta-analyses were conducted to derive pooled risk ratio estimates. 50 studies totaling 14,661,595 people were included. In all populations combined, there was an increased risk of a wide range of 39 out of 40 symptoms in those infected with SARS-CoV-2 compared to uninfected controls. The symptoms with the highest pooled relative risks were loss of smell (RR 4.31; 95% CI 2.66, 6.99), loss of taste (RR 3.71; 95% CI 2.22, 7.26), poor concentration (RR 2.68; 95% CI 1.66, 4.33), impaired memory (RR 2.53; 95% CI 1.82, 3.52), and hair loss/alopecia (RR 2.38; 95% CI 1.69, 3.33). This evidence synthesis, of 50 controlled studies with a cumulative participant count exceeding 14 million people, highlights a significant risk of diverse long-term symptoms in individuals infected with SARS-CoV-2, especially among those who were hospitalised.

It is now well-established that a significant proportion of people who become infected with SARS-CoV-2 go on to experience prolonged and, in some cases, debilitating symptoms for many months following recovery from the initial acute infection, which is commonly referred to as Long Covid or Post-acute COVID-19 syndrome. A 2022 pooled observational analysis of 54 global studies (1.2 million individuals from 22 countries) found 3 months after infection, 6.2% reported 1 of 3 Long Covid symptom clusters, including ongoing respiratory problems (3.7%), persistent fatigue with bodily pain or mood swings (3.2%), and cognitive issues (2.2%), adjusted for pre-existing health[1]. A large systematic review of 194 global studies published in early 2023 reported that, at an average follow-up time of 4 months, 45% of COVID-19 survivors, regardless of hospitalisation status, experienced at least one unresolved symptom[2]. In addition, fatigue, disturbed sleep, and

breathlessness were highly prevalent symptoms reported across hospitalised, non-hospitalised, and mixed cohorts[2]. Over twenty systematic reviews on the prevalence, incidence and long-term health effects of Long Covid have been conducted[2–23]. A major limitation of these reviews is that they have largely synthesised observational studies that have not utilised a comparator or control population. This is especially important when evaluating symptoms prevalent in the general population (e.g., headache, fatigue, sleep problems), or those possibly worsened by the COVID-19 pandemic. The only systematic review in adults that examined prevalence in studies including uninfected controls reported that infection with SARS-CoV-2 carried significantly higher risk of fatigue (risk ratio, RR 1.72, 95% CI: 1.41, 2.10), shortness of breath (RR 2.60, 95% CI: 1.96, 3.44), memory difficulties (RR 2.53, 95% CI: 1.30, 4.93), and concentration difficulties (RR 2.14, 95% CI: 1.25, 3.67) at ≥ 4 weeks following infection[23]. This review of 33 studies however only considered fatigue, shortness of breath, cognitive dysfunction, and quality of life outcomes, and it included studies of healthcare workers, so cannot be considered a review of general populations.

Long Covid has negatively impacted the lives of millions of people globally, and continues to increase the burden on health and social care systems. As a result, it is important to continually evaluate the existing literature on the long-term health effects of Long Covid in the general population to inform health and social care planning, as well as future interventions and therapeutics. The primary objective of this study was to conduct a systematic review and meta-analysis of the risk of all reported symptoms of Long Covid in general populations infected with SARS-CoV-2 compared to uninfected comparator/control populations.

## Results

A total of 48,104 records were screened, with 2495 retrieved for full-text evaluation. A total of 57 cohorts from 50 studies were included and the study flow is reported in a PRISMA flowchart (Fig. 1), with a list of included studies, study characteristics, and references detailed in Supplementary Table 3.

### Characteristics of included studies

Fifty studies totaling 14,661,595 people were included (controls $n = 13,247,625$ and COVID-19 $n = 1,413,970$), with thirteen studies conducted in those <18 years of age. Most studies were conducted in Europe ($n = 25$), Asia ($n = 13$) or North America ($n = 9$) the remaining studies were conducted in Africa ($n = 1$), South America ($n = 1$) or across multiple continents ($n = 1$). The time to follow-up ranged from an average of 28 to 685 days. Eleven studies reported data on hospitalised patients, 14 non-hospitalised, and 25 on hospitalised and non-hospitalised combined (mixed). The most reported symptom outcomes across all cohorts were fatigue ($n = 34$), headache/migraine ($n = 36$), breathlessness ($n = 33$), chest pain/ tightness ($n = 27$), and affected sleep ($n = 25$). Of note, for 35 (70%) of the 50 studies included, ethnicity of the populations was not reported.

The overall quality rating of included studies was low risk of bias ($n = 35$) and medium risk of bis ($n = 15$); see Supplementary Table 4. Most of the studies ($n = 33$) utilised self-report (i.e., no reference to original medical records or laboratory reports to confirm the outcome) to select SARS-CoV-2 negative comparator populations, 15 used record linkage (e.g., identified through ICD codes on database records) and two studies failed to provide a description.

Publication bias was not found to be a statistically significant for the outcome 'one or more symptoms at follow –up', with p-values from the Egger's test of 0.125, 0.266 and 0.282 for the sub-groups of hospitalised, non-hospitalised, and mixed respectively.

### All cohorts irrespective of hospitalisation status

Across all studies combined, a total of 40 individual symptoms were reported (Fig. 2), and the relative risk exceeded 1.0 in all symptoms except for hoarse voice. The five symptoms with the highest pooled relative risks were loss of smell (RR 4.31; 95% CI 2.66, 6.99), loss of taste (RR 3.71; 95% CI 2.22, 7.26), poor concentration (RR 2.68; 95% CI 1.66, 4.33), impaired memory (RR 2.53; 95% CI 1.82, 3.52), and hair loss/alopecia (RR 2.38; 95% CI 1.69, 3.33). The outcome of at least one symptom at follow-up was reported in 17 studies.

### Hospitalised cohorts

Among hospitalised patients, a total of 37 symptoms were reported, and the relative risk exceeded 1.0 for all symptoms except diarrhoea and abdominal pain. The five symptoms with the highest pooled relative risks were loss of smell (RR 11.05; 95% CI 3.02, 40.48), loss of taste (RR 8.59; 95% CI 3.58, 20.60), chills/shivers (RR 7.44; 95% CI 2.14, 25.89), sore throat (RR 4.77; 95% CI 1.30, 17.50), and pain (RR 4.46; 95% CI 3.38, 5.87). The outcome of at least one symptom at follow-up was only reported in six studies. See Fig. S1 for all symptom risk ratio estimates in hospitalised study populations.

### Non-Hospitalised cohorts

In the non-hospitalised groups, a total of 39 symptoms were reported, and the relative risk exceeded 1.0 for all symptoms except weight loss, sore throat, hoarse voice, and cognitive dysfunction. The five symptoms with the highest pooled relative risks were poor concentration (RR 4.86; 95% CI 2.67, 8.85), loss of smell (RR 4.28; 95% CI 2.92, 6.28), hair loss/alopecia (RR 3.09; 95% CI 1.51, 6.33), impaired memory (RR 2.74; 95% CI 1.02, 7.39), and loss of taste (RR 2.22; 95% CI 1.56, 3.17). The outcome of at least one symptom at follow-up was only reported in six studies. See Fig. S2 for all symptom risk ratio estimates in non-hospitalised study populations.

### Mixed Hospitalised and Non-Hospitalised cohorts

In the mixed group of 25 hospitalised and non-hospitalised cohorts, a total of 42 symptoms were reported, and the relative risk exceeded 1.0 for all symptoms except chills/shivers and hoarse voice. The five symptoms with the highest pooled relative risks were indigestion/heart burn (RR 6.35; 95% CI 0.35, 114.17), loss of smell (RR 4.16; 95% CI 2.23, 7.79), impaired memory (RR 3.62; 95% CI 1.85, 7.07), confusion (RR 3.38; 95% CI 2.21, 5.18), and taste (RR 3.27; 95% CI 1.71, 6.27). The outcome of at least one symptom at follow-up was only reported in five studies. See Fig. S3 for all symptom risk ratio estimates in the mixed group of hospitalised and non-hospitalised populations.

### Associations of study characteristics with risk ratio estimates

For many of the meta-analysis models fitted between study heterogeneity was high. Meta-regression models were fitted to assess associations between study level characteristics and estimated effect sizes (Supplementary Table 5). The percentage of male participants was significantly associated with fever, cough, and hair loss; whereby studies with a greater percentage of males showed a greater impact of COVID-19 on these symptoms (higher estimated relative risks). No other significant associations were observed.

## Discussion

We report on the relative risk of Long COVID symptoms in a general population post-COVID-19 compared to uninfected controls using 50 studies including 14,661,595 people. This systematic review found that in all populations combined, there is an increased risk of up to 42 symptoms in those infected with SARS-CoV-2 compared to uninfected controls. Across all populations combined, the greatest increased risk was observed for loss of smell (4.31-fold), loss of taste (3.71-fold), poor concentration (2.68-fold), impaired memory (2.53-fold), and hair loss/alopecia (2.38-fold). Smell and taste disturbances were in the top five symptoms with the highest risk ratio across all three cohorts. Poor concentration, impaired memory, and hair loss/alopecia were in the the top five symptoms with the

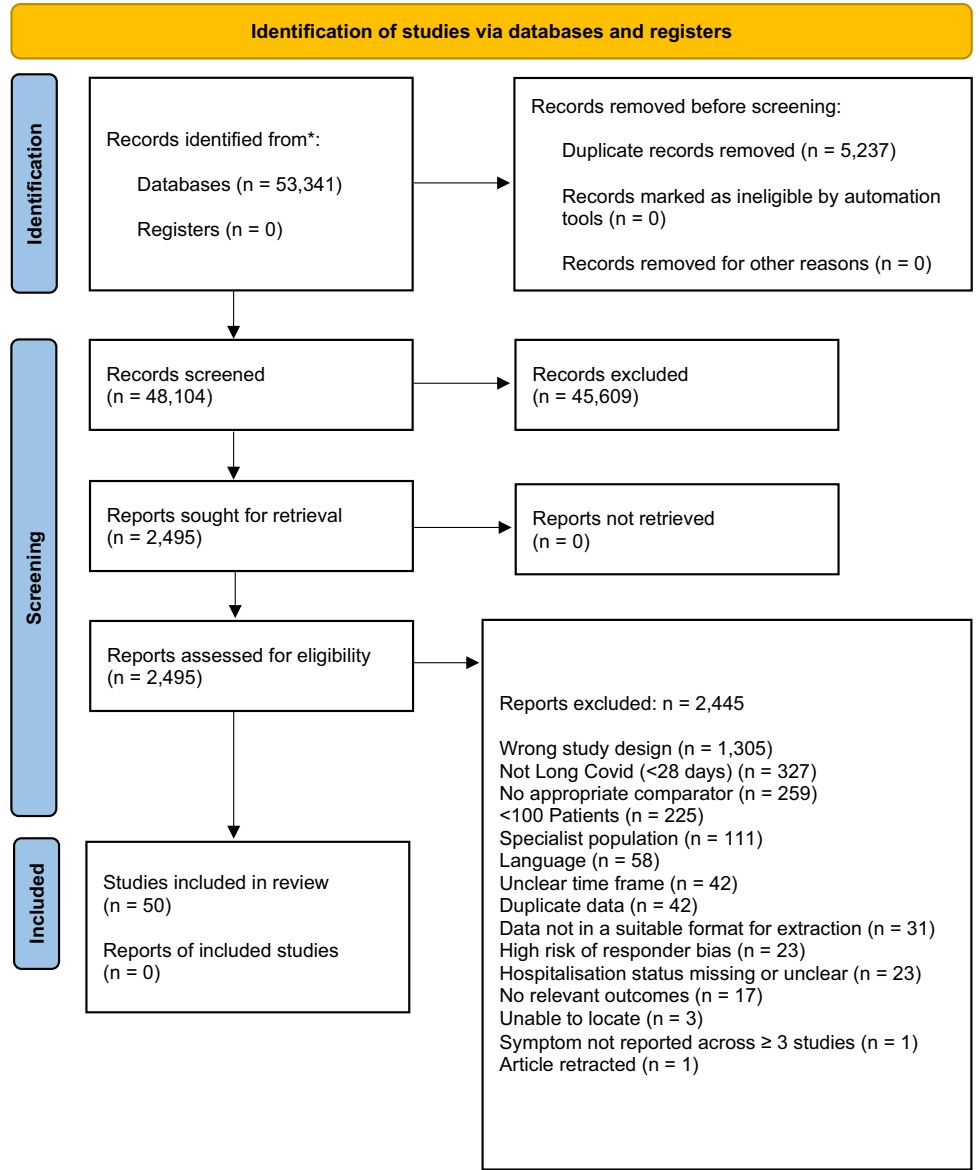

**Fig. 1 | PRISMA flow diagram.** This diagram shows the systematic process we followed to include papers captured by our search, 48,104 records were screened and 50 studies included in the final review.

highest risk ratio across all populations combined and the non-hospitalised cohorts.

In general, risk ratios for symptoms were slightly higher in hospitalised populations, in comparison to non-hospitalised, or mixed hospitalised and non-hospitalised populations. In all cohorts, and in non-hospitalised cohorts, risk ratios were below 1 for hoarse voice, suggesting it is unlikely to be related to SARS-CoV-2 infection. A prior systematic review of controlled studies[23] (which included studies of healthcare workers and veterans) reporting on the risk of main Long Covid symptoms, found that COVID-19 infection resulted in an elevated risk of fatigue (1.72-fold), shortness of breath (2.60-fold), memory problems (1.44-fold), and concentration problems (2.53-fold) at four or more weeks post-infection across all populations combined. In agreement, our data show elevated risk for these symptoms in all populations combined, with lower risk estimates for all except memory (2.53-fold) and concentration (2.68).

In analyses of only hospitalised or outpatient populations, Marjenberg and colleagues[23] showed a similar risk of fatigue and shortness of breath compared with data from all SARS-CoV-2 infections, whereas in the present review, risk ratios were greater in hospitalised compared to non-hospitalised populations.

Similar to Xu and colleagues[24] we observed an increased risk of long-term symptoms in cases compared to controls across a plethora of symptoms, however the risks reported in the current review were often lower. This is likely due to our review being truly reflective of the general population and excluding studies including healthcare workers, veterans and symptomatic controls. Additionally, Xu and colleagues report odds ratios which compared to risk ratios are known to often overestimate risk[25].

A significant limitation of evidence synthesis on Long Covid symptoms has been the lack of comparator and control groups. This is a particularly important requirement when assessing symptoms which are common in the general population (e.g., headache, fatigue, sleep problems)[26], or which may have been exacerbated by living through the COVID-19 pandemic. The present systematic review, in concert with the findings of Marjenberg et al.,[23] clearly demonstrate that SARS-CoV-2 infection, regardless of hospitalization status, is associated with markedly increased risks of a range of long-term symptoms.

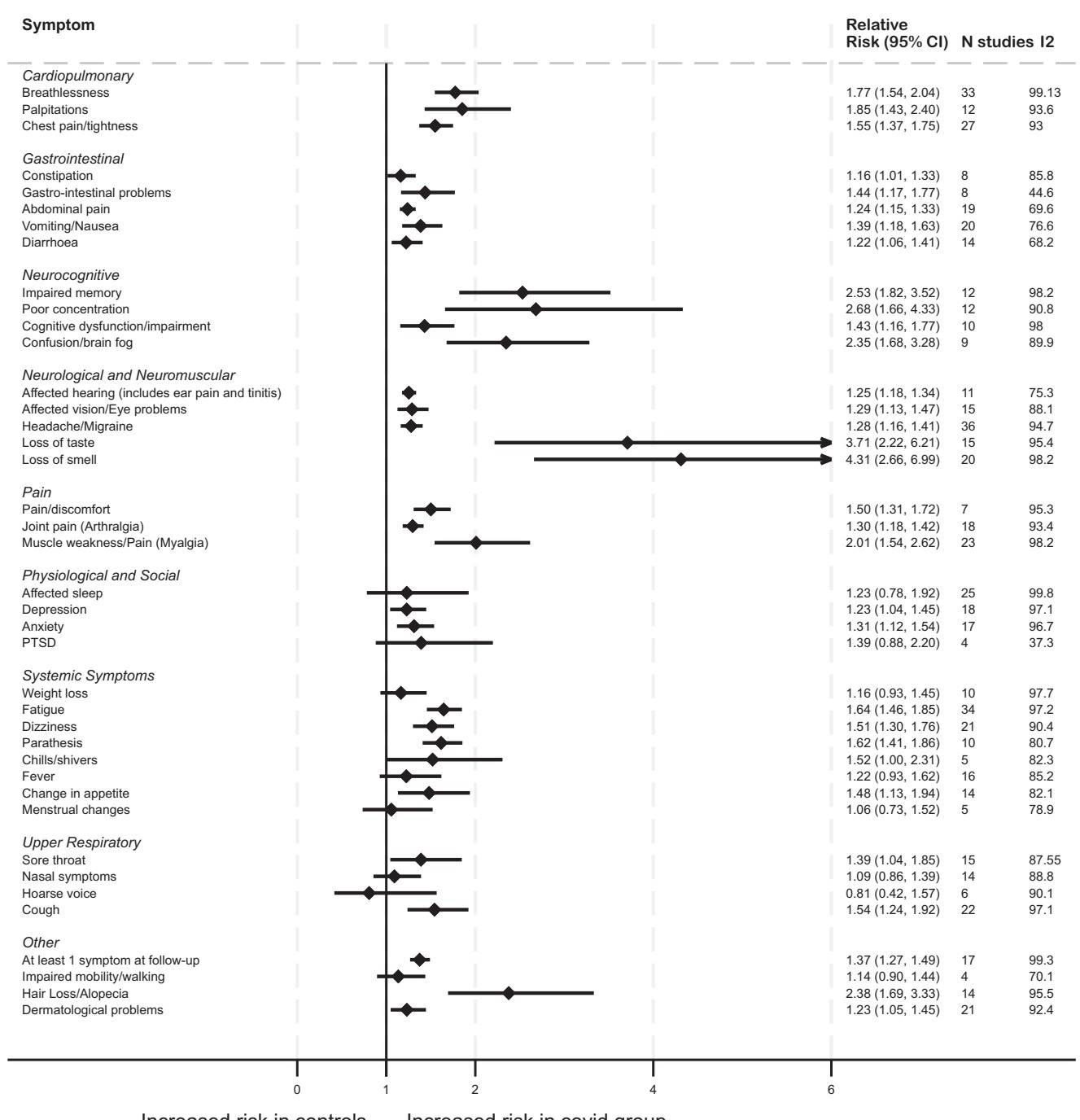

**Fig. 2 | Forest plot of pooled relative risks for each symptom in all populations combined.** The figure illustrates pooled relative risks and associated 95% confidence intervals for each symptom using random effects models. All tests were 2-sided and no adjustment was made for multiple comparisons. Heterogeneity was assessed using Higgins $I^2$ statistic ($I^2$ between 75% and 100% indicates considerable between study heterogeneity).

A previous systematic review of 194 uncontrolled studies on the prevalence of Long Covid symptoms[2], found that fatigue, disturbed sleep, and breathlessness were highly prevalent symptoms reported across hospitalised, non-hospitalised, and mixed cohorts. In the present review, while all displayed risk ratios greater than 1.0, across all populations analysed, none of these symptoms were in the top five symptoms with the highest risk ratio, in any of the cohorts. This demonstrates that identifying the most discriminatory symptom profiles for Long Covid, is highly dependent on the inclusion of a control/comparator group, as well as the varied symptom measures employed, and populations studied.

In the present review, 42 individual symptoms were analysed, which we report across nine symptom groups. A consensus exercise with a multidisciplinary international group previously recommended a core outcome set of 11 outcomes for Long Covid research[27]. However, there has been limited consideration in the literature, core outcome sets, or in the present review, of the total number of reported symptoms as an outcome. Previous reports suggest that the total number of symptoms is positively associated with the severity of ongoing health impairments in adults hospitalised with COVID-19[28] and in children and young people the number of symptoms was more indicative of Long Covid than individual symptoms[29,30]. Without this indicator it is hard to

ascertain the full burden of ill health that people with Long Covid are living with. Data on the total number of symptoms and their combined impact, which was not routinely reported in the included studies, may also help shape the intensity or complexity of care required[28].

Inclusion criteria for the exposure group in the present review were defined as SARS-CoV-2 infection and the presence of ongoing symptoms for a minimum of 28 days but we did not stipulate symptoms must have occurred within a given timeframe of the initial infection. Although the World Health Organization (WHO) definition of Long Covid includes the development of new symptoms[31], recent repeated longitudinal data on children and young people found that test-positive and test-negative reported new symptoms at 6 months[32]. This is problematic when considering the causal relation to the original SARS-COV-2 infection and it is plausible that some new symptoms are due to undiagnosed re-infection with SARS-CoV-2 or simply reflect the background symptomatology found in the general population[33]. In the WHO definition for children and young people, symptoms have to have occurred within 3 months of acute COVID-19[34]. Future evidence reviews should consider utilizing such criteria that consider the issue of new symptom reporting.

Long Covid has been hypothesised to result from various mechanisms such as immune dysregulation, microbiota dysbiosis, autoimmunity and immune pruning, blood clotting and endothelial abnormalities, and dysfunctional neurological signalling, indicating a complex interplay needing further research for targeted treatments.

A limitation of the results presented here, are that many of the meta-analyses showed high between study heterogeneity ($I^2 > 75\%$). This indicates that study estimates are differing due to study level characteristics, such as the populations being studied and this was investigated where possible with meta-regression and sub-group analyses. A final limitation is that although the majority of included studies utilised PCR testing, serology or negative test records via EHR to determine the absence of SARS-CoV-2, many were published in 2022/23 when there may have been few truly negative controls in the UK (and potentially Europe and North America), particularly as access to free testing kits had been withdrawn. However, we did ensure that symptomatic control groups were excluded.

There are various strengths to the current review however. We present the most comprehensive systematic synthesis of global evidence on the risk of symptoms of Long Covid in general populations infected with SARS-CoV-2 compared to uninfected comparator/control populations. In addition, the review was conducted and reported according to the PRISMA guidelines and pre-registered on PROSPERO. Several limitations are worth highlighting however. First, there was geographical homogeneity in the included studies, with 68% of studies being conducted in Europe or North America. There is a continuing need for evidence from regions with a higher proportion of low-and middle-income countries such as Africa, South America, as well as a need for data across a range of ethnic groups and deprivation groups.

Secondly, there was significant diversity in the populations examined, the methodologies employed in the studies, and the outcomes related to symptoms. All included studies followed a cohort design, even though many of them described their design as a case-control study. Additionally, the choice of a non-infected comparator or control population varied among the studies. Thirdly, we did not examine the distinct impact of different SARS-CoV-2 variants or vaccination. We also did not study recover rates from these symptoms, though the longest follow-up period included was 685 days. Investigating the long-term health effects of COVID-19 is however challenging due to the diversity of SARS-CoV-2 variants, their initial severities, potential variations in symptom trajectories, and the influence of widespread vaccination across large segments of populations over time[2]. Finally, we did not incorporate data on diagnoses (e.g.,

myocarditis, pulmonary fibrosis) which would provide further insight into the wider sequelae of Long Covid.

In summary, this review presents relative risk estimates for the long-term health impact assessments of Long Covid in studies of both hospitalised and non-hospitalised groups that included a comparator population. Future studies should include standardising symptom data collection tools for enhancing the clinical relevance of Long Covid reviews. We found that individuals infected with SARS-CoV-2 face an elevated risk of up to 42 symptoms compared to uninfected controls. As a consequence, healthcare services and policies must prioritise Long Covid care and understand its sub-types for targeted healthcare strategies.

## Methods

This review adhered to the Preferred Reporting Items for Systematic Review and Meta-analyses (PRISMA) guidelines, as outlined in the supplemental file (Supplementary Table 1). Additionally, it was pre-registered on the International Prospective Register of Systematic Reviews (PROSPERO; CRD42021238247).

### Inclusion criteria

**COVID-19 exposed population.** To meet the inclusion criteria, studies needed to involve a minimum of 100 individuals with a history of COVID-19, confirmed either through self-diagnosis or by a polymerase chain reaction (PCR), antigen, or antibody test, and who continued to experience symptoms for ≥28 days. The definition of Long Covid, characterised by persistent symptoms lasting ≥28 days, was consistent with our earlier review[2] and aligned with national data on Long Covid recorded by the UK Office for National Statistics and the UK NHS and National Institute for Health and Care Excellence, covering "ongoing symptoms of COVID" over a period of 4 to 12 weeks.

As the focus of this review was on evaluating the risk of Long Covid symptoms in the general population following COVID-19, studies focusing on specific sub-groups such as specialist respiratory clinics, healthcare workers, pregnant women, veterans were excluded. Additionally, studies that failed to report if patients were hospitalised or not, those where the duration of follow-up could not be determined, case studies, and those where all participants were not assessed for a minimum of 28 days, were excluded.

**Study design.** Primary research studies of any design (except case studies) that included a control/comparator group and reported at least one relevant outcome at ≥28 days were deemed eligible. Reviews, editorial, and letters that did not contain original research were excluded.

**Comparator/control population.** Inclusion criteria required studies to incorporate a control or comparator group comprising individuals (with no specified limit on group size) who had not experienced COVID-19, confirmed through self-diagnosis or objective tests such as PCR, antigen, or antibody tests. Symptomatic control groups were excluded and coded as wrong comparator.

**Outcomes.** All included studies were required to present the frequency of at least one symptom or clinical investigation. Included studies reported individual symptom data in the following forms: total number of participants, risk ratios, hazard ratios, or weighted percentages. Other forms of data that could not be used for direct comparisons within the meta-analysis model were excluded (e.g., odds ratios, mean difference from baseline etc.,). When symptoms were combined or clustered, we were unable to use them in the analyses. Studies reporting on serology, histopathology, and clinical biomarkers were beyond the scope of this review. When cohort data is presented at multiple time points (e.g., 3 months and 6 months) data was extracted from the longest follow up point.

## Database searches and screening

We conducted updated searches from our prior systematic review and meta-analysis[2], extending from January 1, 2022, to August 1, 2024. The databases searched included MEDLINE, The Cochrane Library, Scopus, CINAHL, and medRxiv. All records were managed in reference management software EndNote 21 (Philadelphia, United States). Details regarding the search terms and strategy can be located in Table 2 of the supplementary material.

Two reviewers independently evaluated titles, abstracts, and full-text articles identified through database searches against the eligibility criteria using the online collaborative software Covidence (Melbourne, Australia). Any discrepancies were resolved through consensus by a third reviewer.

## Data extraction and risk of bias assessment

A single reviewer carried out the data extraction using a pre-established data extraction form, while a second reviewer independently verified the accuracy of 10% of extractions. Extracted data encompassed study details, population demographics, pertinent outcomes, and COVID-19 status (e.g., hospitalisation status, time to follow-up). To assess the risk of bias of included studies, an adjusted version of the Newcastle-Ottawa Scale was used[23]. Each study received an overall risk of bias rating of low, medium or high. Two reviewers independently assessed risk of bias using a pre-established data extraction form. Any discrepancies were resolved through consensus discussion. Details on the risk of bias can be found in Table S4 of the supplementary material.

## Data synthesis

Due to the multi-system nature of Long Covid, we have presented relative risk estimates for symptom data by hospitalisation status and by the following systems/groupings: (i) systemic, (ii) pain, (iii) cardiopulmonary, (iv) gastrointestinal, (v) upper respiratory, (vi) neurological and neuromuscular, (vii) physiological and social, (viii) neurocognitive, and (ix) other; see Fig. 2 and Supplementary Figs. 1–3. We detail outcomes reported in five or more studies across hospitalised, non-hospitalised, and mixed cohorts.

Where relative risks or hazard ratios were reported within identified studies, these were extracted for use within the meta-analyses. Where these were not reported, relative risks were computed for each study using reported numbers. Pooled relative risks with 95% CI were estimated for each symptom using random effects models. Heterogeneity was assessed using Higgins $I^2$ statistic[35]. Meta-analyses were carried out for all symptoms that were reported across 3 or more studies. Due to study populations falling into three broad categories, meta-analyses were carried out for all studies, and then by sub-groups based on hospital status (hospitalised, non-hospitalised or mixed). Where information from 4 or more studies was available, between study heterogeneity was investigated by fitting meta-regression models to assess the associations between study effect size with mean study age of patients (median age was used if mean was not reported) and sex (proportion of males). The potential for publication bias was evaluated within meta-analyses by funnel plots and the Egger's test[36]. All analyses were carried out in Stata/IC 18.0.

## Reporting summary

Further information on research design is available in the Nature Portfolio Reporting Summary linked to this article.

## Data availability

The dataset generated during and analysed during the current study are available in the Figshare repository (https://doi.org/10.6084/m9.figshare.28695185.v1).

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

## Acknowledgements

This work is funded by National Institute for Health and Care Research Applied Research Collaboration East Midlands NIHR ARC East Midlands). K.K. is supported by the National Institute for Health Research (NIHR) Applied Research Collaboration East Midlands (ARC EM) and the NIHR Leicester Biomedical Research Centre (BRC). S.J. is supported by the National Institute for Health Research (NIHR) Applied Research Collaboration East Midlands (ARC EM). We would also like to thank academic librarian Keith Nockels for his assistance with the searches. This review was part funded by PHOSP-COVID. PHOSP-COVID is jointly funded by a grant from the MRC-UK Research and Innovation and the Department of Health and Social Care through the National Institute for Health Research (NIHR) rapid response panel to tackle COVID-19 (grant references: MR/V027859/1 and COV0319). The views expressed in the publication are those of the author(s) and not necessarily those of the National Health Service (NHS), the NIHR or the Department of Health and Social Care.

## Author contributions

K.K., L.L.O. and A.R. conceptualised the study. L.L.O., A.R., S.A.J., A.A., S.C., W.E., M.P.F., A.W., A.P., S.J., A.P.K., T.J.W. and A.W. performed data curation. CG carried out the statistical analyses and accessed and verified the underlying study data. A.R. wrote the first draft of the manuscript. All authors acquired the data. L.L.O., A.R., C.G., D.A., A.B., C.B., R.A.E. M.C., M.P., S.S., R.S., T.S., J.S., H.W., T.W., K.K. interpreted findings. All authors critically revised the manuscript for intellectual content. All authors had full access to the data in the study and had final responsibility for the decision to submit for publication.

## Competing interests

A.B. is PI of the NIHR-funded STIMULATE-ICP study (COV-LT2-0043) and has also received other research funding from Astra Zeneca, NIHR, BMA, UKRI and EU. M.C. is Director of the Birmingham Health Partners Centre for Regulatory Science and Innovation, Director of the Centre for Patient Reported Outcomes Research and is a National Institute for Health Research (NIHR) Senior Investigator. MC receives funding from the National Institute for Health Research (NIHR), UK Research and Innovation (UKRI), NIHR Birmingham Biomedical Research Centre, the NIHR Surgical Reconstruction and Microbiology Research Centre, NIHR ARC West Midlands, UK SPINE, European Regional Development Fund – Demand Hub and Health Data Research UK at the University of Birmingham and University Hospitals Birmingham NHS Foundation Trust, Innovate UK (part of UKRI), Macmillan Cancer Support, UCB Pharma, Janssen, GSK and Gilead. She is the senior author of the Symptom Burden Questionnaire™ for Long COVID. M.C. has received personal fees from Aparito Ltd, CIS Oncology, Takeda, Merck, Daiichi Sankyo, Glaukos, GSK and the Patient-Centered Outcomes Research Institute (PCORI) outside the submitted work. In addition, a family member owns shares in GSK. All research at Great Ormond Street Hospital NHS Foundation Trust and UCL Great Ormond Street Institute of Child Health is made possible by the NIHR Great Ormond Street Hospital Biomedical Research Centre. The views expressed are those of the author(s) and not necessarily those of the NHS, the NIHR or the Department of Health (R.S. and T.S.). T.S. is Chair of the Health Research Authority for England and recused himself from all Research Ethics Applications. H.W. is an NIHR Senior Investigator Award and acknowledges support from the NIHR Imperial Biomedical Research Centre, Health Data Research UK, NIHR Applied Research Collaborative North West London, and the Wellcome Trust. K.K. is Chair of the Ethnicity Subgroup of the UK Scientific Advisory Group for Emergencies (SAGE) and Member of SAGE and also Chair of the National Long Covid Research Working Group which informs the Chief Medical Officer. K.K. has acted as a consultant, speaker or received grants for investigator-initiated studies for Astra Zeneca, Sanofi-Aventis, Pfizer and Roche. The remaining authors declare no competing interests.

## Additional information

[1]Diabetes Research Centre, University of Leicester, Leicester, UK. [2]Leicester Real World Evidence Unit, Diabetes Research Centre, University of Leicester, Leicester, UK. [3]School of Psychology and Vision Sciences, University of Leicester, Leicester, UK. [4]Department of Family and Community Medicine, College of Medicine, King Faisal University, Al Hofuf, Saudi Arabia. [5]Office for National Statistics, Government Buildings, Newport, UK. [6]Faculty of Population Health Sciences, Institute of Health Informatics, University College London, London, UK. [7]Department of Population Science and Experimental Medicine, University College London, London, UK. [8]Department of Respiratory Sciences, University of Leicester, Leicester, UK. [9]Department of Applied Health Sciences, University of Birmingham, Birmingham, UK. [10]NIHR Birmingham Biomedical Research Centre and NIHR Applied Research Collaboration West Midlands, University Hospital Birmingham and University of Birmingham, Birmingham, UK. [11]Birmingham Health Partners Centre for Regulatory Science and Innovation and Centre for Patient Reported Outcomes Research, University of Birmingham, Birmingham, UK. [12]School of Social Sciences and Humanities, Aston University, Birmingham, UK. [13]NIHR Leicester Biomedical Research Centre, Respiratory Department, University Hospitals of Leicester NHS Trust, Leicester, UK. [14]School of Sport, Exercise and Health Sciences, Loughborough University, Loughborough, UK. [15]Development Centre for Population Health, University of Leicester, Leicester, UK. [16]School of Public Health, University College Cork, Cork, Ireland. [17]Great Ormond Street Institute of Child Health, University College London, London, UK. [18]Population Health Sciences, Bristol Medical School, University of Bristol, Bristol, UK. [19]Faculty of Medicine, School of Public Health, Imperial College London, London, UK. [20]These authors contributed equally: Lauren L. O'Mahoney, Ash Routen. ✉e-mail: kk22@leicester.ac.uk

