## [Transparent Peer Review file · Nature Communications]

The risk of Long Covid symptoms: a systematic review and meta-analysis of controlled studies.

Corresponding Author: Dr Lauren O'Mahoney

Version 0:

Reviewer comments:

Reviewer #1

(Remarks to the Author)

The authors conducted a systematic review and meta-analysis on long-Covid. It compares populations infected with SARS-CoV-2 with uninfected populations. There are some issues that should be addressed.

First, the authors stated that 'A major limitation of these reviews is that they have largely synthesized observational studies that have not utilized a comparator or control population. This is especially important when evaluating symptoms prevalent in the general population (e.g., headache, fatigue, sleep problems), or those possibly worsened by the COVID-19 pandemic.' However, I am afraid this reason alone is not sufficient enough to distinguish the current study to the many other reviews. I would like to see the authors dig into the importance of their study.

Second, in this systematic review and meta-analysis, the search of the literature was made to March 2023. This has been almost one and a half years. Since this is a hot topic with an increasing number of studies, it is important for the authors to update their search and analysis. Also, it is possible to add results on the absolute scales together with raw numbers in supplementary materials.

Third, in terms of the pooled results, it may be possible to calculate crude RR or RD and then run the analysis with meta-regression. This can be done and compared with the available results. This is important because an important purpose of meta-analysis is to identify patterns of reported findings rather than generate a summary estimate. I would like to see the authors further dig into patterns of reported findings.

Fourth, for the risk of bias assessment, it is unclear to me why only 30% of included papers were independently assessed by a second reviewer. There were only 36 studies included in the review, so it should not be challenging to have two reviewers rate these studies separately.

Fifth, I would like the authors to have a more extensive discussion of their findings, especially considering that the majority of the populations have been infected at least by one variant to date. If over 80% or 90% have been previously infected, what is the implication of making the comparison between populations with and without infection?

Reviewer #2

(Remarks to the Author)

The authors did not address my concerns satisfactorily. I remain concerned about the lack of novelty of this analysis, its narrow scope on "symptoms" and the fact that it is not comprehensive.

a. The authors argue that this is the first systematic review of controlled studies. This is not accurate. There are several -- including this one: <https://www.nature.com/articles/s41598-023-42321-9> and many others.

b. None of the findings reported are novel. The authors indicate that the novelty stems from their assessment that

SARSCoV-2 infection, regardless of hospitalisation status, is associated with markedly increased risks of a range of long-term symptoms. This has been known since 2020 and is widely reported in the literature including numerous systematic reviews.

c. I was concerned about the exclusive focus on "symptoms". Instead of broadening the scope of the review to include other health effects -- which will make the more useful and impactful, the authors chose to include the narrow focus on symptoms as a limitation. It is fine. it just undermines the impact of the work and makes it far less interesting.

d. I was concerned about lack of data on vaccination. The authors again opted to add a statement in the discussion instead of actually adding studies that include vaccination status (there are many).

e. it is disappointing that "the comprehensive search strategy with a specialist academic librarian" missed the study that I provide them earlier. Instead of including it in the meta-regression, they opted to only include it in the introduction. A huge missed opportunity to integrate it along with the other data. I understand that it is easier to just write a sentence in the introduction than redo the analyses, but this lessens the impact dramatically. Also, there are other papers that are missing. Clearly, the "comprehensive search strategy" employed by the authors missed some. I am tempted to provide the authors with the list of missing references, but i will refrain from doing so. It is not my job to do the search for them. I already pointed out that their "comprehensive search" is not indeed comprehensive, and supplied one missing study as an example. The rest is up to them if they chose to actually be comprehensive.

Version 1:

Reviewer comments:

Reviewer #1

(Remarks to the Author)

Thanks to the authors for addressing most of my concerns except for one issue:

I mentioned that 'Fourth, for the risk of bias assessment, it is unclear to me why only 30% of included papers were independently assessed by a second reviewer. There were only 36 studies included in the review, so it should not be challenging to have two reviewers rate these studies separately.'

The authors responded 'The decision to have a second reviewer assess 30% of the studies was made to balance resource constraints while still ensuring rigor in the review process. A single reviewer assessed risk of bias using a pre-established data extraction form, while a second reviewer independently scored 30% of included papers. The percentage agreement for individual items on the Newcastle-Ottawa Scale between reviewers was 81%.'

This means that around 7 studies included may have inconsistent quality assessment if the remaining studies were assessed by a second reviewer. This is a big number out of 50 included studies. Therefore, this has to be addressed. Besides, in the author list, there are over 10 researchers, so I think this should not be an outstanding issue.

Reviewer #2

(Remarks to the Author)

The authors addressed my concerns.

Reviewer Comment	Author Response	Location of Change
Reviewer 1		
The authors conducted a systematic review and meta-analysis on long-Covid. It compares populations infected with SARS-CoV-2 with uninfected populations. There are some issues that should be addressed. First, the authors stated that ‘A major limitation of these reviews is that they have largely synthesized observational studies that have not utilized a comparator or control population. This is especially important when evaluating symptoms prevalent in the general population (e.g., headache, fatigue, sleep problems), or those possibly worsened by the COVID-19 pandemic.’ However, I am afraid this reason alone is not sufficient enough to distinguish the current study to the many other reviews. I would like to see the authors dig into the importance of their study.	To the best of our knowledge, this review represents the first attempt to provide relative risk estimates for the long-term health impacts of Long Covid across studies involving both hospitalized and non-hospitalized general populations, all of which included non-symptomatic comparator or control groups. As noted in the introduction, there have been over 20 systematic reviews addressing the prevalence, incidence, and long-term health effects of Long Covid. While some of these reviews included comparator/control groups, they were limited in scope. For example, one review (PMID: 37714919, PMCID: PMC10504382) focused only on fatigue, shortness of breath, cognitive dysfunction, and quality of life outcomes, and included studies on healthcare workers, which makes it unsuitable as a review of the general population. Similarly, another recent systematic review (PMCID: PMC11317913, PMID: 39129538) also included a mix of populations, including healthcare workers and veterans, thus again failing to provide a comprehensive review of the general population. This underscores the distinction of our work, as we provide a focused assessment of Long Covid’s long-term health impacts on the broader general population and a wide range of symptoms, specifically incorporating studies that include control/comparator groups.	N/A
Second, in this systematic review and meta-analysis, the search of the literature was made to March 2023. This has been almost one and a half years. Since this is a hot topic with an increasing number of studies, it is important for the authors to update their search and analysis.	We have now updated our search (developed with a specialist academic librarian) to August 1st 2024 and included new papers and conducted an updated analysis.	Figure 1 (Page 23)

Also, it is possible to add results on the absolute scales together with raw numbers in supplementary materials.	From March 2, 2023 to August 1, 2024 the updated search identified 14,212 studies of which 14 additional studies were included in the review. Our results section is already extensive, so we made the decision not to present both RR and RDs. Raw numbers are given in the supplementary table if absolute differences need to be considered by the reader.	
Third, in terms of the pooled results, it may be possible to calculate crude RR or RD and then run the analysis with meta-regression. This can be done and compared with the available results. This is important because an important purpose of meta-analysis is to identify patterns of reported findings rather than generate a summary estimate. I would like to see the authors further dig into patterns of reported findings.	Where data was available a number of meta-regressions were carried out and these are reported in Supplementary Table 5. Due to the small numbers of studies once we split by symptom and setting, the scope for meta-regressions was limited.	N/A
Fourth, for the risk of bias assessment, it is unclear to me why only 30% of included papers were independently assessed by a second reviewer. There were only 36 studies included in the review, so it should not be challenging to have two reviewers rate these studies separately.	We appreciate your concern regarding the risk of bias assessment. The decision to have a second reviewer assess 30% of the studies was made to balance resource constraints while still ensuring rigor in the review process. A single reviewer assessed risk of bias using a pre-established data extraction form, while a second reviewer independently scored 30% of included papers. The percentage agreement for individual items on the Newcastle-Ottawa Scale between reviewers was 81%. Any discrepancies were resolved through consensus discussion.	N/A
Fifth, I would like the authors to have a more extensive discussion of their findings, especially considering that the majority of the populations have been infected at least by one variant to date. If over 80% or 90% have been previously infected, what is the implication of making the comparison between populations with and without infection?	Our Discussion has been extended to include comparison with another recent systematic review titled 'Excess risks of long COVID symptoms compared with identical symptoms in the general population: A systematic review and meta-analysis of studies with control groups' (PMCID: PMC11317913, PMID: 39129538). Although the authors state this review was in the general population it included health care	Lines 363-369 and 426-431.

	workers, veterans, and symptomatic control groups. Further, in our Discussion we acknowledge the following regarding control/comparator populations having been infected. “A final limitation is that although the majority of included studies utilised PCR testing, serology or negative test records via EHR to determine the absence of SARS-CoV-2, many were published in 2022/23 when there may have been few truly negative controls in the UK (and potentially Europe and North America), particularly as access to free testing kits had been withdrawn. However, we did ensure that symptomatic control groups were excluded.”	
Reviewer 2		
The authors did not address my concerns satisfactorily. I remain concerned about the lack of novelty of this analysis, its narrow scope on "symptoms" and the fact that it is not comprehensive. a. The authors argue that this is the first systematic review of controlled studies. This is not accurate. There are several -- including this one: https://www.nature.com/articles/s41598-023-42321-9 and many others.	To the best of our knowledge, this review is the first to provide relative risk estimates for the long-term health impacts of Long Covid across studies involving both hospitalized and non-hospitalized general populations, all of which include comparator or control groups, not just some. As mentioned in the introduction, over 20 systematic reviews have examined the prevalence, incidence, and long-term health effects of Long Covid. While some of these reviews did include comparator/control groups, they were limited in scope. For instance, the review by Marjenberg (PMID: 37714919, PMCID: PMC10504382) focused only on fatigue, shortness of breath, cognitive dysfunction, and quality of life outcomes, and included studies on healthcare workers, making it unsuitable as a review of the general population. Additionally, databases were only searched up to 23rd March 2022. Whereas the new review now includes studies up to August 1st, 2024. Similarly, another recent review (PMCID: PMC11317913, PMID: 39129538) also	N/A

	mixed populations, including healthcare workers, which again limits its applicability to the general population. This highlights the distinction of our work, as we provide a focused assessment of Long Covid's long-term health impacts on the broader general population and a broad range of symptoms, specifically incorporating studies with control/comparator groups.	
b. None of the findings reported are novel. The authors indicate that the novelty stems from their assessment that SARSCoV-2 infection, regardless of hospitalisation status, is associated with markedly increased risks of a range of long-term symptoms. This has been known since 2020 and is widely reported in the literature including numerous systematic reviews.	We agree that the association between SARS-CoV-2 infection and long-term symptoms has been widely replicated since 2020. However, the absence of control groups has been reported to have led to distortion of risk (https://doi.org/10.1136/bmjebm-2023-112338). The findings from previous reviews have been limited due to the omission of control groups and subsequent lack of information on pre-existing symptom prevalence in the general population where symptoms may already have been high before the pandemic. Therefore, it was important to determine if the risk of symptoms remained high when compared to the general population who had not been infected. Our systematic review reports important novel finding as we have taken into account pre-existing symptom prevalence in the general population.	N/A
c. I was concerned about the exclusive focus on "symptoms". Instead of broadening the scope of the review to include other health effects -- which will make the more useful and impactful, the authors chose to include the narrow focus on symptoms as a limitation. It is fine. it just undermines the impact of the work and makes it far less interesting.	As we stated in our previous response, we acknowledge this limitation of the review, which is addressed in the Discussion with the following statement: "Finally, we did not incorporate data on diagnoses (e.g., myocarditis, pulmonary fibrosis) which would provide further insight into the wider sequelae of Long Covid." However, majority of Long Covid diagnosis have been based on ongoing symptoms which was the aim of this	N/A

	systematic review.	
d. I was concerned about lack of data on vaccination. The authors again opted to add a statement in the discussion instead of actually adding studies that include vaccination status (there are many).	We agree that the impact of vaccination and era/variant is important. However, as acknowledged in our previous response, the majority of studies included in our review do not provide data on vaccination status or the specific variants of SARS-CoV-2. Additionally, these studies were conducted across numerous countries, where the dominance of variants likely differed over time. In our view, this variability prevents the meaningful division of studies into sub-cohorts based on era or variant. As noted in the Discussion section, we explicitly address this issue with the following statement: “Thirdly, we did not examine the distinct impact of different SARS-CoV-2 variants or vaccination. We also did not study recovery rates from these symptoms, and the longest follow-up period included was a little over 12 months. Investigating the long-term health effects of COVID-19 is, however, challenging due to the diverse nature of SARS-CoV-2 variants, their initial severities, potential variations in symptom trajectories, and the influence of widespread vaccination across large segments of populations over time.” We feel this acknowledges the complexities involved in studying these factors.	N/A
e. it is disappointing that "the comprehensive search strategy with a specialist academic librarian" missed the study that I provide them earlier. Instead of including it in the meta-regression, they opted to only include it in the introduction. A huge missed opportunity to integrate it along with the other data. I understand that it is easier to just write a sentence in the introduction than redo the analyses, but this lessens the impact dramatically. Also, there are other papers	We have now updated our search, in collaboration with a specialist academic librarian, to include studies up to August 1st, 2024. This update incorporates new papers, and we have conducted an updated analysis. Regarding the previous paper you provided (PMID: 36215063, PMCID: PMC9552043, DOI: 10.1001/jama.2022.18931) it did not	Lines 177-180 and 191-196

that are missing. Clearly, the "comprehensive search strategy" employed by the authors missed some. I am tempted to provide the authors with the list of missing references, but I will refrain from doing so. It is not my job to do the search for them. I already pointed out that their "comprehensive search" is not indeed comprehensive, and supplied one missing study as an example. The rest is up to them if they chose to actually be comprehensive.

meet our inclusion criteria and was excluded due to 'wrong study design'.

The study included 54 studies in total. Of the 54 studies, 44 were previously published and 10 were collaborating cohorts.

The reason systematic reviews are always excluded from new reviews is to ensure studies are not 'double counted' as this can bias and distort findings. Many of the original research studies included in the Global Burden of Disease study are already included in our review (e.g., CloCK cohort; Xiong et al. 2021; Huang et al., 2022) and therefore including the Global Burden of Disease study would be highly inappropriate. Furthermore, data for the 10 collaborating cohorts was not reported separately and symptoms were reported as clusters and not individually. We have added the following statements to provide clarity to the reader.

"Primary research studies of any design (except case studies) that included a control/comparator group and reported at least one relevant outcome at ≥ 28 days were deemed eligible. Reviews, editorial, and letters that did not contain original research were excluded."

"Included studies reported individual symptom data in the following forms: total number of participants, risk ratios, hazard ratios, or weighted percentages. Other forms of data that could not be used for direct comparisons within the meta-analysis model were excluded (e.g., odds ratios, mean difference from baseline etc.). When symptoms were combined or clustered, we were unable to use them in the analyses."

As we mentioned in our previous response, we did reference the Global Burden of Disease study in the

	introduction and discussion sections, using the following statement: “A 2022 pooled observational analysis of 54 global studies (1.2 million individuals from 22 countries) found that 3 months after infection, 6.2% reported one of three Long Covid symptom clusters, including ongoing respiratory problems (3.7%), persistent fatigue with bodily pain or mood swings (3.2%), and cognitive issues (2.2%), adjusted for pre-existing health.” This highlights our inclusion of the study within the broader context of Long Covid research and we trust we have explained why the study itself has not been included in the current review.	
--	---	--

Additional Comments

Four new authors (SJ, MPF, AW, and AP) have been added to the author list during our revisions. All new authors were added due to data curation/screening and contribution to the final manuscript, and therefore meet the journals criteria for authorship.

Reviewer Comment	Author Response	Location of Change
Reviewer 1		
Thanks to the authors for addressing most of my concerns except for one issue: I mentioned that 'Fourth, for the risk of bias assessment, it is unclear to me why only 30% of included papers were independently assessed by a second reviewer. There were only 36 studies included in the review, so it should not be challenging to have two reviewers rate these studies separately.' The authors responded 'The decision to have a second reviewer assess 30% of the studies was made to balance resource constraints while still ensuring rigor in the review process. A single reviewer assessed risk of bias using a pre-established data extraction form, while a second reviewer independently scored 30% of included papers. The percentage agreement for individual items on the Newcastle-Ottawa Scale between reviewers was 81%.' This means that around 7 studies included may have inconsistent quality assessment if the remaining studies were assessed by a second reviewer. This is a big number out of 50 included studies. Therefore, this has to be addressed. Besides, in the author list, there are over 10 researchers, so I think this should not be an outstanding issue.	Thank you for your feedback. We have now addressed your concerns and the risk of bias for all studies has been assessed independently by two reviewers. Any discrepancies were resolved through consensus discussion. We have provided evidence of this in the form of an excel spreadsheet. We have updated the manuscript and supplementary material to reflect the changes following this process.	Page 10 (Lines 217 – 222) Page 12 (Lines 267- 272) Supplementary Material (Table S4)